# Solving Reflected Diffusion Models: A PINNs-Based Method

## Abstract

Score-based reflected diffusion models generate approximations of high-dimensional data distributions while respecting the known constraints of the data distribution by learning a reversed reflected stochastic differential equation evolving within the support of the data. Similar to standard diffusion models, the theoretical convergence of reflected diffusion models is based on bounded errors of score estimations. However, the existence and attainability of low-error score estimators have not yet been studied in the reflected diffusion setting. In this paper, we construct a novel score estimator using the Physics-Informed Neural Network (PINN), solving reflected diffusion models in a deep-learning fashion. We proceed to derive a uniform theoretical error bound of $O(N^{-\frac{1}{4}})$ for the score function on a training dataset of sample size $N$ at any time $t \in [0, T]$ of the diffusion process. This result fills the gap between theory and practice in the score estimation of the reflected diffusion model. Moreover, its independence of dimension ensures the performance of our estimator in large-sample scenarios under high-dimensional settings.

## 1 Introduction

Diffusion models are a powerful family of deep generative models that generate high-quality approximation samples for high-dimensional samples, achieving state-of-the-art performance in various applications Sohl-Dickstein et al. (2015); Song & Ermon (2019); Ho et al. (2020); Song et al. (2020). The effectiveness of diffusion models allows it to be widely used in text and image generation Ho et al. (2020); Dhariwal & Nichol (2021); Nichol et al. (2021); Ramesh et al. (2022), text-to-image synthesis Jeong et al. (2021); Popov et al. (2021); Huang et al. (2022), audio generation, and video generation Schneider (2023); Gupta et al. (2025). It has also been proven useful in numerous extended scientific fields such as biostatistics, neuroscience, and material science, where high-quality generative modeling is required, covering problems such as molecular generation Xu et al. (2023); Luo et al. (2024) and protein folding Yi et al. (2023); Huang et al. (2024).

Traditional diffusion models map data to Gaussian noise, which does not respect the possible boundary constraints of the original data; this could result in large errors in the tail probability and generate highly unnatural samples. To address the mismatch between the data domain and the diffusion domain, recent studies have brought to the attention of reflected diffusion models Lou & Ermon (2023). Reflected diffusion models are an extension of traditional diffusion models that incorporate boundary conditions. It employs stochastic processes that evolve within a bounded domain, with reflections at the boundary to ensure the process remains within the domain. These models are particularly useful in applications where the data or process being modeled is confined to a bounded region, such as physical systems, finance, or constrained optimization problems Fishman et al. (2023). Lou & Ermon (2023) have shown that reflected diffusion models outperform traditional diffusion models theoretically and empirically to generate data constrained within a bounded domain.

Closely related to diffusion models, Score-based generative modeling is a powerful framework for generative modeling that has gained significant attention in recent years. Score-based generative modeling Song & Ermon (2019); Song et al. (2020) focuses on learning the score function of a data distribution, which is the gradient of the log-probability density function. Once the score function is learned, samples can be generated from a known prior distribution by reversing a forward process that reconstructs the data distribution with Gaussian noise, using iterative methods such as Langevin dynamics. The approximation of the score function is typically achieved by minimizing the loss of

score match Hyvärinen (2005) by training neural networks, as seen in works such as Bond-Taylor et al. (2021); Chen et al.. Diffusion models can be viewed as a specific instance of score-based generative modeling. Song et al. (2021) unified score-based models and diffusion models under the framework of SDEs. They showed that both approaches can be described as discretizations of continuous-time SDEs, providing a deeper theoretical understanding of their connections.

While score-based diffusion models outperform empirically by achieving state-of-the-art results in a variety of fields, there are still unanswered questions regarding the theoretical guarantee of score estimation. Prior works employ Girsanov's Theorem and Pinsker's Inequality to bound the total variation distance between the true and approximated reverse diffusion process Chen et al. (2022); Chen et al.; Benton et al. (2024), which can be similarly applied to the reflected diffusion setting Lou & Ermon (2023). However, most of the statistical estimation error bound results of the score-based diffusion models and reflected diffusion models require the assumption that there exists a score estimator that uniformly-in-time minimizes the score estimation error Chen et al.; Benton et al. (2024), while leaving the theoretical guarantee of the existence of such a minimizer undiscussed. Works such as Zhang et al. (2024) fill this gap for the score function in the form of $p_0 \star \mathcal{N}(0, t\mathbf{I}_d)$, which corresponds to the diffusion model with an Ornstein-Uhlenbeck process. However, it isn't easy to promote this result to more general diffusion models, as well as reflected diffusion models. In particular, there is still a lack of discussion of theoretical guarantees of the score estimator in the reflected diffusion setting.

To fill this theory gap and provide a score-estimation methodology for reflected diffusion models with theoretical guarantees, we introduce the PINNs-score estimator. Physics-Informed Neural Networks (PINNs) are a framework that combines neural networks with physical laws, typically expressed as PDEs, to solve scientific and engineering problems Raissi et al. (2019). PINNs leverage the universal approximation capabilities of neural networks while enforcing physical constraints, making them powerful tools for solving forward and inverse PDE problems. By using a PINNs-based algorithm to solve the Fokker-Planck Equation for the density function of the forward reflected diffusion process, we build a novel score estimator for the reflected diffusion models. Moreover, using the theoretical convergence properties of PINNs derived by Grönwall's inequality, we obtain a uniform-in-time score estimation error for reflected diffusion models under mild assumptions.

## 1.1 MAIN CONTRIBUTIONS

**Novelty of Method.** The solution of diffusion models has been tightly connected with the solution of PDEs by the Fokker-Planck equation derived from the famous Feynman-Kac formula of Stochastic Differential Equations. There has been some research on using PDE and PINN methods to estimate generative models (Berner et al. (2022); Máté & Fleuret; Sun et al. (2024)) over the last few years. However, those works mainly focus on solving the log-density PDE of diffusion models in $\mathbb{R}^d$, where the instability of data can often happen. In addition, although numerical analyses in those works have shown that the PINN method can outperform traditional sampling methods such as Path Integral Sampler (Zhang & Chen) and Schrodinger Bridge, there hasn't been a thorough theoretical error analysis regarding this topic. In this paper, we take advantage of the connection between SDEs and PDEs in a bounded and regular domain and novelly employ PINNs in the solution of **reflected diffusion models**, which makes better use of the boundary constraint training feature of PINNs, avoiding the instability of error when the tail probability is high, and making our theoretical analysis natural and reasonable.

**Theoretical Guarantees For Score Estimators.** For the score function $s_t(x) = \nabla \log p_t(x)$ of a reflected diffusion model for $d-$ dimension data in the bounded domain $\Omega = [0, 1]^d$ with diffusion time $t \in [0, T]$, we construct an estimator $\hat{s}_t(x) = \nabla \log \hat{p}_t(x)$, where $\hat{p}_t(x)$ is the estimation of the probability density function $p_t(x)$ of the forward process. We state that $\hat{p}_t(x)$ can be constructed via PINNs by solving a Fokker-Planck partial differential equation with Neumann boundary conditions. We proceed to derive a uniform-in-time theoretical error bound of $O(N^{-\frac{1}{4}})$ for the score function at any time $t \in [0, T]$ of the finite diffusion process. This is the first score estimation method in the setting of reflected diffusion models with a specific theoretical error-bound guarantee.

**Broad Applicability.** Our result only requires mild assumptions on the drift term and diffusion term of the forward diffusion process, and thus can be applied to various kinds of reflected diffusion models, including diffusions with more complex coefficients than simple Brownian motion or the Ornstein-

Uhlenbeck process. In particular, we derived the specific requirements of the diffusion coefficients for our error analysis in the assumptions of our main theorem. Requiring solely smoothness and lower boundedness of the original data distribution, our result can also be used in a broad variety of data generation problems, including generating high-dimensional data in bounded domains.

## 1.2 RELATED WORKS

**Score-based Diffusion Models.** Score-based generative models are introduced based on prior works on generative modeling, notably Denoising Diffusion Probabilistic Models (DDPMs), which were introduced by Sohl-Dickstein et al. (2015) and later refined by Ho et al. (2020). DDPMs define a forward process that gradually adds Gaussian noise to data and a reverse process that learns to denoise it. The forward process is fixed, while the reverse process is learned using a neural network. Song and Ermon Song & Ermon (2019); Song et al. (2020) introduced score-based generative models, which learn the gradient of the data distribution (the score function) and use Langevin dynamics to generate samples. These models are closely related to diffusion models and were later unified under the framework of Stochastic Differential Equations (SDEs) by Song et al. (2021). For a comprehensive survey of diffusion models, see Yang et al. (2023).

**Reflected Diffusion Models.** The theoretical background of reflected diffusion models is stochastic differential equations with reflecting boundary conditions, which is studied in Lions & Sznitman (1984); Tanaka (1979); Costantini (1992). Lou & Ermon (2023) systemizes the idea of reflected diffusion models and solves it via a score-matching technique. They also demonstrated the superiority of reflected diffusion models over traditional diffusion models with thresholding techniques in preventing the generation of highly unnatural samples in bounded data settings.

**Physics-Informed Neural Networks.** Before the advent of PINNs, solving PDEs mostly relies on traditional numerical methods, such as Finite Element Methods (FEM), Finite Difference Methods (FDM), and Finite Volume Methods (FVM). However, these methods often fall short with high-dimensional continuous-domain problems. The key motivation of Physics-Informed Neural Networks was to leverage the universal approximation capabilities of neural networks and their ability to handle high-dimensional data, using neural networks as function approximators for PDE solutions. The modern formulation of PINNs was introduced by Raissi et al. (2019). The potential of PINNs is demonstrated by its application in various fields. For example, Jin et al. (2021) uses PINNs to solve the Navier-Stokes equations for fluid flow, including laminar and turbulent flows in fluid dynamics. It is also used in heat transfer Cai et al. (2021), structural mechanics Bastek & Kochmann (2023), and biomedical engineering Banerjee et al. (2024). With its early applications, the theoretical analysis of PINNs has also advanced, with works such as Mishra & Molinaro (2021; 2022); Shin et al. (2020); De Ryck et al. (2023); Liu et al. (2025) establishing theoretical foundations for PINNs, including convergence properties and error bounds.

## 1.3 BASIC SETTINGS, SYMBOLS AND NOTATIONS

Without loss of generality, this paper studies reflected diffusion models on the $d-$dimensional unit cube $\Omega = [0,1]^d$. We denote $p_{\text{data}} = p_0$ as the original target data distribution, which is also the start of the forward process. All the data points like $x_t$ are $d-$dimensional vectors. We denote $\mathbb{R}_+$ as the set of non-negative real numbers, $\mathcal{N}(D)$ and $\mathcal{U}(D)$ as the standard normal distribution and uniform distribution on a domain $D$, "$\cdot$" as the dot product of vectors, and $\mathbb{E}[\mathbf{X}]$ as the expectation of random variable $\mathbf{X}$.

## 2 BACKGROUND

### 2.1 DIFFUSION MODELS AND SCORE MATCHING

Diffusion models are a family of generative models that reconstruct the original data by injecting noise and then use the reversed denoising process to generate data.

In the following discussion, we only consider the case where the distribution of the original data admits a density function. Under this prerequisite and some necessary regularity conditions, all of the probability distributions appearing in the discussion have a density function. For the sake of

simplicity, for a data distribution with probability measure $p$, we abuse the notation and also denote its density function as $p$.

### 2.1.1 FORWARD AND BACKWARD PROCESS

Given a target data distribution with density $p_{\text{data}}$, the forward process $(\mathbf{X}_t)_{t \in [0,T]}$ of a diffusion model is defined as the solution of the stochastic diffusion equation (SDE) with the initial distribution following $p_{\text{data}}$:

$$\mathrm{d}\mathbf{X}_t = \mathbf{f}(t, \mathbf{X}_t)\mathrm{d}t + g(t)\mathrm{d}\mathbf{B}_t, \quad \mathbf{X}_0 \sim p_{\text{data}}, \tag{1}$$

where $\mathbf{f} : [0, T] \times \mathbb{R}^d \to \mathbb{R}^d$, a vector-valued function whose output's dimension matches the data's dimension, is called "drift term" of the process, and $g : [0, T] \to \mathbb{R}$ is called "diffusion term" of the process. $(\mathbf{B}_t)_{t \in [0,T]}$ is a $d$-dimensional standard Brownian motion which serves as the "noise" added to the data distribution. The forward process is performed on the finite time interval $[0, T]$.

Denote the probability density function of $\mathbf{X}_t$ by $p_t$. In order to generate new samples following the distribution $p_{\text{data}}$, we need to reverse the diffusion process equation 1 in time, which yields the backward process

$$\mathrm{d}\mathbf{Y}_t = (-\mathbf{f}(T - t, \mathbf{Y}_t) + g^2(T - t)\nabla \log p_{T-t}(\mathbf{Y}_t))\mathrm{d}t + g(T - t)\mathrm{d}\mathbf{B}_t, \quad \mathbf{Y}_0 \sim p_T, \tag{2}$$

where $(\mathbf{B}_t)_{t \in [0,T]}$ is another independent $d$-dimensional standard Brownian motion. (We identify the notation of all Brownian motions for simplicity.)

As the solution of the above backward SDE, $(\mathbf{Y}_t)_{t \in [0,T]}$ satisfies that $\mathbf{Y}_t = \mathbf{X}_{T-t}$ in distribution. As such, learning the backward process equation 2 and sampling $\mathbf{Y}_0$ from a prior distribution $p_T = p_{\text{prior}}$, we can generate samples of the desired data distribution as $\mathbf{Y_T} = \mathbf{X}_0 \sim p_{\text{data}}$.

### 2.1.2 SCORE-MATCHING TECHNIQUE

To solve the backward SDE equation 2, the estimation of the function $\nabla \log p_{T-t}$ is needed, which brings out the concept of the score function. Given a probability density function $p(x) : \mathbb{R}^d \to \mathbb{R}_+$, its score function is defined as $s(x) = \nabla_x \log p(x)$. By constructing an estimator $\hat{s}_t(x)$ for the score function $s(x)$ for each $t \in [0, T]$, new samples of data can be generated from the score-based diffusion model

$$\mathrm{d}\hat{\mathbf{Y}}_t = (-\mathbf{f}(T - t, \hat{\mathbf{Y}}_t) + g^2(T - t)\hat{s}_{T-t}(\hat{\mathbf{Y}}_t))\mathrm{d}t + g(T - t)\mathrm{d}\mathbf{B}_t, \quad \hat{\mathbf{Y}}_0 \sim p_{\text{prior}}. \tag{3}$$

The score-matching technique estimates the score function $\nabla \log p_t$ by minimizing its $L^2$-loss over a function class $\mathcal{F}$ for each $t \in [0, T]$,

$$\hat{s}_t = \operatorname{argmin}_{f_t \in \mathcal{F}} \mathbb{E}_{x \sim p_t}[\|f_t(x) - \nabla \log p_t(x)\|^2],$$

where $\mathcal{F}$ is usually a class of neural networks. While the $L^2$-loss function is the most widely used loss function for score-matching, the existence of a uniform-in-time minimizer $\hat{s}_t$ for all $t \in [0, T]$ is not theoretically guaranteed in general cases. Zhang et al. (2024)

### 2.2 REFLECTED DIFFUSION MODELS

More often than not, the target data distribution has a bounded support on $\mathbb{R}^d$. However, the standard diffusion model projects the data into the unbounded domain $\mathbb{R}^d$ by adding stochastic noise throughout the forward and backward process, which could compound the error and result in the generation of highly unnatural samples Song & Ermon (2019); Koehler et al. (2023). Many diffusion models adapt the thresholding technique Ho et al. (2020); Li et al. (2022) to solve this problem, but this method leaves a gap between theory and practice due to the mismatch between training and generative process. Ho & Salimans (2021); Saharia et al. (2022); Lou & Ermon (2023)

To address this problem, reflected diffusion models Lou & Ermon (2023) are broadly used for data with bounded constraints. For data distributions inside a bounded domain $\Omega \in \mathbb{R}^d$, like the standard diffusion models, reflected diffusion models generate data samples by the forward and backward process, but under this case, the forward process $(\mathbf{X}_t)_{t \in [0,T]}$ becomes the solution of a reflected SDE

$$\mathrm{d}\mathbf{X}_t = \mathbf{f}(t, \mathbf{X}_t)\mathrm{d}t + g(t)\mathrm{d}\mathbf{B}_t + \mathrm{d}\mathbf{L}_t, \quad \mathbf{X}_0 \sim p_{\text{data}}, \tag{4}$$

where $(\mathbf{L}_t)_{t \in [0,T]}$ is the additional stochastic constraint which forces $\mathbf{X}_t$ to stay inside $\Omega$ by reflecting $\mathbf{X}_t$ once it hits the boundary $\partial\Omega$. Respectively, the backward process is the solution of the reversed reflected SDE Cattiaux (1988); Williams (1988):

$$
\begin{aligned}
\mathrm{d}\mathbf{Y}_t =& (-\mathbf{f}(T - t, \mathbf{Y}_t) + g^2(T - t)\nabla \log p_{T-t}(\mathbf{Y}_t))\mathrm{d}t \\
& + g(T - t)\mathrm{d}\mathbf{B}_t + \mathrm{d}\bar{\mathbf{L}}_t, \quad \mathbf{Y}_0 \sim p_T,
\end{aligned}
\tag{5}
$$

where $(\bar{\mathbf{L}}_t)_{t \in [0,T]}$ is the reversed stochastic boundary condition.

A thorough introduction of reflected SDEs can be found in Pilipenko (2014). Particularly, the reflected SDE has a unique strong solution if $\mathbf{f}$ and $g$ are Lipschitz in state and time, and $\Omega$ is sufficiently regular, which holds under this paper's assumption $\Omega = [0,1]^d$.

## 2.3 PHYSICS-INFORMED NEURAL NETWORKS

Physics-Informed Neural Network (PINNs) is a practical and efficient algorithm to solve forward and inverse problems for PDEs. For a general form of an abstract PDE,

$$
\begin{aligned}
\mathcal{L}[u](x,t) &= f(x,t), \quad \forall x \in \Omega, \quad \forall t \in [0,T], \\
\mathcal{B}_k[u](x,t) &= h_k(x,t), \quad \forall x \in \Gamma_k \subset \partial\Omega, \\
\forall t &\in [0,T], \quad k = 1, \cdots, K, \\
u(x,0) &= \phi(x), \quad \forall x \in \Omega
\end{aligned}
$$

the loss function of PINNs is constructed based on minimizing the pointwise residuals:

$$
\begin{aligned}
L(\theta)^2 =& \int_{\Omega \times [0,T]} |\mathcal{L}[u_\theta] - f|^2 \mathrm{d}x\mathrm{d}t \\
& + \sum_{k=1}^{K} \int_{\Gamma_k \times [0,T]} |\mathcal{B}_k[u_\theta] - h_k|^2 \mathrm{d}s(x)\mathrm{d}t \\
& + \int_{\Omega} |u_\theta(x,0) - \phi(x)|^2 \mathrm{d}x.
\end{aligned}
\tag{6}
$$

In practice, the integrals in Equation (6) are approximated by suitable numerical quadratures, which gives the training loss

$$
\begin{aligned}
L(\theta)^2 =& \sum_{n=1}^{N_l} \omega_l^n |\mathcal{L}[u_\theta](x_l^n, t_l^n) - f(x_l^n, t_l^n)|^2 \\
& + \sum_{k=1}^{K} \sum_{n_k=1}^{N_b^k} \omega_b^{n_k} |\mathcal{B}_k[u_\theta](x_b^{n_k}, t_b^{n_k}) - h_k(x_b^{n_k}, t_b^{n_k})|^2 \\
& + \sum_{n=1}^{N_0} \omega_0^n |u_\theta(x_0^n, 0) - \phi(x_0^n)|^2 \mathrm{d}x.
\end{aligned}
$$

## 3 MAIN RESULTS

### 3.1 PINNS-ESTIMATION OF THE FOKKER-PLANCK EQUATION

To use PINNs to solve reflected diffusion problems, we need to convert the reflected SDE problem equation 4 to a PDE problem, which can be derived from the Feynman-Kac formula.

**Theorem 3.1.** *Lou & Ermon (2023) Given the (strong) solution* $(\mathbf{X}_t)_{t \in [0,T]}$ *of the forward reflected SDE equation 4, let* $p_t$ *denote the probability density function of* $\mathbf{X}_t$ *for* $t \in [0,T]$. *For* $\Omega = [0,1]^d$, *let* $u(x,t) = p_t(x)$, *then the function* $u : [0,1]^d \times [0,T] \to \mathbb{R}_+$ *satisfies the Fokker Planck Equation*

*(FPE) with Neumann Boundary Condition Schuss (2013):*

$$\partial_t u = \nabla \cdot (-u\mathbf{f} + \frac{g^2}{2}\nabla u),$$

$$u(\cdot, 0) = p_0 = p_{data},$$

$$\left(u(x, t)\mathbf{f} - \frac{g^2}{2}\nabla u\right) \cdot \mathbf{n} = 0, \forall x \in \partial\Omega, t \in (0, T].$$

*Remark* 3.2. From the theorem above, it's not hard to see that when the diffusion coefficients $\mathbf{f}$ and $g$ satisfy the following conditions:

1. $\mathbf{f}(\mathbf{x}, \mathbf{t})$ converges to some $\mathbf{f}_0(x)$ uniformly on $\Omega$ as $t \to \infty$.

2. $\lim_{t\to\infty} \frac{g(t)^2}{2} = g_0 > 0$.

3. $\int_\Omega \exp\left(\sum_{i=1}^d \mathbf{f}_0^{(j)}(x)\right)\mathrm{d}x < \infty$.

Then 4 has a stationary distribution with a density function $p_\infty(x) = Z\exp\left(\frac{\sum_{j=1}^d \mathbf{f}_0^{(j)}(x)}{g_0}\right)$, where

$Z = \left(\int_\Omega \exp(\sum_{j=1}^d \mathbf{f}_0^{(j)}(x)/g_0)\right)^{-1}$. Choosing appropriate drift $\mathbf{f}$ and diffusion $g$, we can compute the stationary distributions of reflected diffusion models using this observation, which make generating sample by the backward process from $p_{\text{prior}} \approx p_\infty$ viable. In particular, when $\Omega$ is a bounded convex set, $\mathbf{f} \equiv 0$, and $g$ is bounded, the stationary distribution is the uniform distribution on $\Omega$.

For a training function $u_\theta \in \mathcal{U}$, where $\mathcal{U}$ is a class of neural networks with smooth activation function (such as $\sigma = \tanh$) and $\theta \in \Theta$ as tuning parameters, define the FPE residuals as follows:

The PDE residual $R_{PDE}$

$$R_{PDE}(\theta, x, t) = \partial_t u_\theta - \nabla \cdot (-u_\theta \mathbf{f} + \frac{g^2}{2}\nabla u_\theta), x \in \Omega, t \in [0, T],$$

the boundary condition residual $R_{bc}$

$$R_{bc}(\theta, x, t) = \left(u_\theta(x, t)\mathbf{f}(x, t) - \frac{g(t)^2}{2}\nabla u_\theta(x, t)\right) \cdot \mathbf{n}, x \in \partial\Omega, t \in (0, T),$$

and the initial condition residual $R_{ic}$

$$R_{ic}(\theta) = \text{KL}(p_{\text{data}}, u_\theta(\cdot, 0)),$$

Here $\text{KL}(p_{\text{data}}, u_\theta(\cdot, 0))$ is the Kullback-Leibler divergence (KL divergence, also known as Relative Entropy) of $u_\theta(\cdot, 0)$ from $p_{\text{data}}$. We can train and solve the Fokker-Planck Equation by constructing the following loss function for PINNs

$$L(\theta) = \frac{1}{T}\int_0^T \int_\Omega |R_{PDE}|^2 p_t(x)\mathrm{d}x\mathrm{d}t + \frac{1}{T}\int_0^T \int_{\partial\Omega} |R_{bc}|^2 \mathrm{d}s(x)\mathrm{d}t + \lambda R_{ic}(\theta), \qquad (7)$$

where $\lambda$ is a training parameter.

In practice, we need to choose appropriate quadratures to approximate the integrals in Equation (7). In the setting of the reflected diffusion models, it is natural to construct the quadratures using samples generated by the forward process equation 4 for training samples.

Based on FDE residuals, we need to divide our training samples into three parts: Interior training set $\{(x_{PDE}^{(k,i)}, t_k^{PDE}) \mid i = 1, \cdots, N_{PDE}; k = 1, \cdots, N_T\}$ for the PDE residual, boundary training set $\{(x_{bc}^{(k,i)}, t_k^{bc}) \mid i = 1, \cdots, N_{bc}; k = 1, \cdots, N_T\}$ for boundary condition residual, and temporal boundary training set $\{x_{tb}^{(i)}\}_{i=1}^{N_{tb}}$ for initial condition residual. Then we can construct the training loss

---

**Algorithm 1** PINNs-Estimation of The Fokker-Planck Equation

---

**Require:** target data distribution $p_{\text{data}}$ from which we can sample training points.
**for** $n = 1$ **to** $N_{\text{training\_step}}$ **do**
    Sample $t_k \sim \mathcal{U}([0, T])$, $k = 1, \cdots, N_T$.
    Sample the temporal boundary training set $\{x_{tb}^{(i)}\}_{i=1}^{N_{tb}} \sim p_{\text{data}}$.
    On $\{t_k\}_{1 \le k \le N_T}$, sample interior training set $\{(x_{PDE}^{(k,i)}, t_k)\}_{i=1}^{N_{PDE}}$ based on $\{x_{tb}^{(i)}\}_{i=1}^{N_{tb}} \sim p_{\text{data}}$ by
    the forward process Equation (4), sample the boundary training set $\{(x_{bc}^{(i)}, t_i^{bc})\}_{i=1}^{N_{bc}} \sim \mathcal{U}(\partial\Omega)$.
    Compute the training loss $\hat{L}_{train}(\theta)$ in Equation (9).
    Optimize network tuning parameter $\theta$ by minimizing $\hat{L}_{train}(\theta)$. (using methods such as gradient
    descent.)
**end for**
**Return:** Parameterized estimation function $u_\theta(x, t)$.

---

function of PINNs as follows:

$$
\begin{aligned}
L_{train}(\theta) =& \frac{1}{N_{PDE}N_T} \sum_{i=1}^{N_{PDE}} \sum_{k=1}^{N_T} |R_{PDE}(\theta, x_{PDE}^{(k,i)}, t_k^{PDE})|^2 \\
&+ \frac{1}{N_{bc}N_T} \sum_{k=1}^{N_T} \sum_{i=1}^{N_{bc}} |R_{bc}(\theta, x_{bc}^{(k,i)}, t_k^{bc})|^2 \\
&+ \lambda \int_\Omega p_{\text{data}}(x) \log(p_{\text{data}}(x)) \mathrm{d}x \\
&- \frac{\lambda}{N_{tb}} \sum_{i=1}^{N_{tb}} \log(u_\theta(x_{tb}^{(i)}, 0)).
\end{aligned}
\tag{8}
$$

Since $\lambda \int_\Omega p_{\text{data}}(x) \log(p_{\text{data}}(x)) \mathrm{d}x$ is a constant independent of $\theta$, minimizing $L_{train}(\theta)$ is equivalent as minimizing

$$
\begin{aligned}
\hat{L}_{train}(\theta) =& \frac{1}{N_{PDE}N_T} \sum_{i=1}^{N_{PDE}} \sum_{k=1}^{N_T} |R_{PDE}(\theta, x_{PDE}^{(k,i)}, t_k^{PDE})|^2 \\
&+ \frac{1}{N_{bc}N_T} \sum_{k=1}^{N_T} \sum_{i=1}^{N_{bc}} |R_{bc}(\theta, x_{bc}^{(k,i)}, t_k^{bc})|^2 \\
&- \frac{\lambda}{N_{tb}} \sum_{i=1}^{N_{tb}} \log(u_\theta(x_{tb}^{(i)}, 0)),
\end{aligned}
\tag{9}
$$

which can be computed directly from empirical samples.

We conclude the sampling and training process as Algorithm 1.

### 3.2 THE PINNs-SCORE ESTIMATOR FOR REFLECTED DIFFUSION MODELS

By solving the Fokker-Planck equation for the forward reflected SDE using PINNs, we obtain an estimation of the probability density function of the forward process $(\mathbf{X}_t)_{t \in [0,T]}$. We can then construct the PINNs-score estimator for reflected diffusion models:

Suppose $\tilde{u}(x, t) = u_{\theta^\star}(x, t)$ is the estimator of $u(x, t) = p_t(x)$ trained by Algorithm 1, then the PINNs-score estimator for reflected diffusion models can be defined as $\hat{s}_t(x) = \nabla_x \log \tilde{u}(x, t)$. Running the backward reflected SDE with the score function replaced by our score estimator

$$
\mathrm{d}\hat{\mathbf{Y}}_t = (-\mathbf{f}(T - t, \mathbf{Y}_t) + g^2(T - t)\hat{s}_{T-t}(\hat{\mathbf{Y}}_t))\mathrm{d}t + g(T - t)\mathrm{d}\mathbf{B}_t + \mathrm{d}\bar{\mathbf{L}}_t, \quad \hat{\mathbf{Y}}_0 \sim p_{\text{prior}}, \tag{10}
$$

New samples of the target data can be generated with the reflected diffusion model.

# 4 THEORETICAL GUARANTEES

In this section, we derive the theoretical error bound for our PINNs score estimator.

## 4.1 ERROR BOUND FOR THE PINNS-SCORE ESTIMATOR

To establish the PINNs-score estimation error bound, we need the following assumptions on the probability density flow $p_t(x)$, the drift term $\mathbf{f}$, the diffusion term $g$ and the training networks.

**Assumption 4.1.** The probability density function $u(x,t) = p_t(x)$ and network training function $u_\theta(x,t)$, $\theta \in \Theta$ are uniformly upper and lower bounded by some positive constants $M$ and $m$. In addition, $u, u_\theta \in H^k([0,T] \times \Omega)$ for some $k \geq 3$, where $H^k(X)$ is the Hilbertian Sobolev space of order $k$ for domain $X$. (see definition in Theorem A.3, Appendix Section.)

*Remark* 4.2. The upper-boundedness and lower-boundedness of the training networks $u_\theta$ can be realized simply by applying truncations on the networks. We can also apply function smoothing to the training networks to meet the requirement $u_\theta \in H^k([0,T] \times \Omega)$.

**Assumption 4.3.** $\mathbf{f}$ and $g$ are continuous, Lipschitz and sufficiently smooth w.r.t. state $x$ and time $t$, and satisfies

$$\|\mathbf{f}\|_\infty := \|\mathbf{f}\|_{L^\infty([0,T] \times \Omega)} < \infty,$$
$$\max_{t \in [0,T]} |g(t)| \leq C_{g,\max} < \infty.$$

**Assumption 4.4.** $\mathbf{f}$ and $g$ satisfies that: for $\forall t \in [0,T]$, $g(t)^2 - \|\mathbf{f}\|_\infty > 0$, and that

$$\delta_0 := \min_{t \in [0,T]} (g(t)^2 - \|\mathbf{f}\|_\infty) > 0.$$

In conclusion, the convergence of the error bound requires the boundedness and smoothness of $u$, $u_\theta$, $\mathbf{f}$, $g$, and their derivatives, which can be satisfied by choosing appropriate drift and diffusion terms for reflected diffusion models, as well as neural network parameters.

We present our main theorem as follows:

**Theorem 4.5.** *(Score-estimation error bound for PINNs-score estimator). Assume all the assumptions above are satisfied, the PINNs-score estimator $\hat{s}_t(x)$ in Section 3.2 satisfies that for $\Omega = [0,1]^d$, $\forall t \in [0,T]$, $\forall \epsilon_0 > 0$, with probability at least $1 - \frac{2N_T}{\sqrt{N_{PDE}}} - \frac{2}{\sqrt{N_{tb}}}$,*

$$\|\hat{s}_t - \nabla \log p_t\|_{L^2([0,t] \times \Omega)} \lesssim C \left( \epsilon_0 + (N_{PDE} + N_{bc} + N_{tb})^{-1/8} N_T^{-1/8} \right),$$

*where $C$ is a constant dependent on $\lambda$, $m$, $M$, $T$, $k$, $\|\mathbf{f}\|_\infty$ and $\delta_0$, independent of sample sizes and dimension $d$.*

*Remark* 4.6. Denote the total sample size as $\tilde{N} = N_{PDE} + N_{bc} + N_{tb} + N_T$, Theorem 4.5 guarantees a uniform-in-time error bound of $O(\tilde{N}^{-\frac{1}{4}})$. It is also worth mentioning that the constant $C$ that controls the convergence rate of this error bound does not rely on the dimension $d$ of the data distribution, making our method more efficient in theory in high-dimensional settings compared to similar results as in for example Zhang et al. (2024).

The proof of Theorem 4.5 uses integration by parts technique and Grönwall's Inequality. The full proof can be found in the Appendix Section.

## 4.2 CONVERGENCE OF THE BACKWARD PROCESS

With the error bound for score estimation, we can derive the error bound in KL divergence for backward processes $(\mathbf{Y}_t)_{t \in [0,T]}$ and $(\hat{\mathbf{Y}}_t)_{t \in [0,T]}$ as in Equation (5) and Equation (10) using Girsanov's Theorem. The theoretical guarantees of this topic have been well-studied by the existing literature, as shown by the theorem below:

**Theorem 4.7.** *(Reflected Girsanov for KL divergence,Lou & Ermon (2023)). Suppose we have two reflected SDEs on the same domain $\Omega$,*

$$\mathrm{d}x_t = \mathbf{f}_1(x_t, t)\mathrm{d}t + g(t)\mathrm{d}\mathbf{B}_t + \mathrm{d}\mathbf{L}_t,$$
$$\mathrm{d}y_t = \mathbf{f}_2(x_t, t)\mathrm{d}t + g(t)\mathrm{d}\mathbf{B}_t + \mathrm{d}\mathbf{L}_t,$$

*from $t = 0$ to $T$ with $x_0 = y_0 = z \in \Omega$.*

*Let $\mu$, $\nu$ be the path measures with respect to $x$ and $y$. Then,*

$$\mathbb{E}_\mu[\log \frac{\mathrm{d}\mu}{\mathrm{d}\nu}] = \frac{1}{2}\int_0^T \mathbb{E}_{p_{x_t}(y)}[g(t)^2 \|\mathbf{f}_1(y, t) - \mathbf{f}_2(y, t)\|^2]\mathrm{d}t. \tag{11}$$

Under our settings, the right-hand side of Equation (11) becomes

$$\frac{1}{2}\int_0^T \mathbb{E}_{x_t \sim p_t}[|\hat{s}_t(x_t) - s_t(x_t)|^2],$$

which can be bounded using Theorem 4.5.

## 5 CONCLUSIONS

In this paper, we employ the powerful PINNs method in deep learning to solve reflected diffusion models. With the problems of PDE-solving and score estimation bridged by the Fokker-Planck equation of the backward diffusion process, we propose the PINNs-score estimator for estimating score-based reflected diffusion models. In comparison with the traditional score-matching technique, which minimizes the $L^2$-loss of the score function, our method guarantees the existence of an optimal minimizer with a uniform-in-time error bound of $O(N^{-\frac{1}{4}})$ for score estimation. Moreover, this result has no dependence on data dimension $d$, which avoids the explosion of error bound when $d$ is large, ensuring the efficiency of our estimator in high-dimensional settings. Our result is the first score estimation method in the setting of reflected diffusion models with a specific theoretical error-bound guarantee, which holds great theoretical significance.

**Ethics Statement.** All the authors of this paper have carefully read the Code of Ethics. This research adheres to the ethical standards set forth by the ICLR community. We affirm that no part of this research was conducted in violation of ethical standards.

**Reproducibility statement.** The main theoretical results for this research are presented in Section 4. A complete proof of the claims can be found in the appendix, along with clear explanations of backgrounds, lemmas and corollaries. All the assumptions are explicitly listed and thoroughly discussed in Section 4.

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

# A APPENDIX

## A.1 AUXILIARY LEMMAS

To prove the theoretical results in Section 4, we need to introduce some auxiliary lemmas first.

**Definition A.1.** (The $\chi^2$-distance, Gibbs & Su (2002)). For measure space $\Omega$, if $f, g$ are densities of the measure $\mu, \nu$ with respect to a dominating measure $\lambda$, and $S(\mu), S(\nu)$ are their supports on $\Omega$, the $\chi^2$-distance between $\mu$ and $\nu$ are defined as

$$d_{\chi^2}(\mu, \nu) := \int_{S(\mu) \cup S(\nu)} \frac{(f-g)^2}{g} \mathrm{d}\lambda.$$

**Lemma A.2.** *(The KL divergence and $\chi^2$-distance, Gibbs & Su (2002)). The KL divergence and $\chi^2$-distance between $\mu$ and $\nu$ on measure space $\Omega$ satisfy*

$$\mathrm{KL}(\mu, \nu) \leq d_{\chi^2}(\mu, \nu).$$

*Proof.* Since $\log$ is a concave function, according to Jensen's inequality,

$$\mathrm{KL}(\mu, \nu) = \int_\Omega \log(f/g) f \mathrm{d}\lambda \leq \log(\int_\Omega (f/g) f \mathrm{d}\lambda).$$

On the other hand,

$$\int_\Omega \frac{(f-g)^2}{g} \mathrm{d}\lambda = \int_\Omega (\frac{f^2}{g} - 2f + g) \mathrm{d}\lambda = \int_\Omega \frac{f^2}{g} \mathrm{d}\lambda - 1,$$

so

$$\mathrm{KL}(\mu, \nu) \leq \log(1 + d_{\chi^2}(\mu, \nu)) \leq d_{\chi^2}(\mu, \nu).$$

$\square$

**Definition A.3.** (Sobolev spaces and Hilbertian Sobolev spaces). Let $d \in \mathbb{N}_+$, $k \in \mathbb{N}$, $1 \leq p \leq \infty$ and let $\Omega \in \mathbb{R}^d$ be open. With $L^p(\Omega)$ the usual Lebesgue measurable space, we define the Sobolev space $W^{k,p}(\Omega)$ as

$$W^{k,p}(\Omega) = \{f \in L^p(\Omega) : D^\alpha f \in L^p(\Omega), \forall \alpha \in \mathbb{N}^d \text{ with } |\alpha| \leq k\}.$$

For $p < \infty$, we define the the following norms on $W^{k,p}(\Omega)$,

$$\|f\|_{W^{k,p}(\Omega)} = \left( \sum_{m=0}^{k} \sum_{|\alpha|=m} \|D^\alpha f\|_{L^p(\Omega)}^p \right)^{1/p},$$

and for $p = \infty$ we define the norm

$$\|f\|_{W^{k,\infty}(\Omega)} = \max_{0 \leq m \leq k, |\alpha|=m} \|D^\alpha f\|_{L^\infty(\Omega)}.$$

The space $W^{k,p}(\Omega)$ equipped with the norm $\|\cdot\|_{W^{k,p}(\Omega)}$ is a Banach space. Define the Hilbertian Sobolev spaces for $k \in \mathbb{N}$ as $H^k(\Omega) = W^{k,2}(\Omega)$ with corresponding norms $\|\cdot\|_{H^k(\Omega)} = \|\cdot\|_{W^{k,2}(\Omega)}$. If $k$ is large enough, the space $H^k(\Omega)$ is a Banach algebra.

For functions in Hilbertian Sobolev spaces, we have the following two properties:

**Lemma A.4.** *For $d, k \in \mathbb{N}_+$ with $k > \frac{d}{2}$, $H^k(\Omega)$ is a Banach algebra i.e., there exists $c_k > 0$ such that for all $u, v \in H^k(\Omega)$,*

$$\|uv\|_{H^k(\Omega)} \leq c_k \|u\|_{H^k(\Omega)} \|v\|_{H^k(\Omega)}.$$

**Lemma A.5.** *(Multiplicative trace inequality, e.g. De Ryck et al. (2023), Lemma A.3). Let $d \geq 2$, $\Omega \in \mathbb{R}^d$, let $\gamma_0 : H^1(\Omega) \to L^2(\partial\Omega) : u \longmapsto u|_{\partial\Omega}$ be the trace operator. Denote by $h_\Omega$ the diameter of $\Omega$ and by $p_\Omega$ the radius of the largest $d$-dimensional ball that can be inscribed into $\Omega$. Then it holds that*

$$\|\gamma_0 u\|_{L^2(\partial\Omega)} \leq \sqrt{\frac{2 \max\{2h_\Omega, d\}}{p_\Omega}} \|u\|_{H^1(\Omega)}.$$

The next lemma shows that functions in the Hilbertian Sobolev spaces can be approximated by tanh neural networks in Sobolev norms.

**Lemma A.6.** *(De Ryck et al. (2023), Theorem B.7). Let $d, n \geq 2$, $m \geq 3$, $\delta > 0$, $a_i, b_i \in \mathbb{Z}$ with $a_i < b_i$ for $1 \leq i \leq d$, $\Omega = \prod_{i=1}^{d} [a_i, b_i]$ and $f \in H^m(\Omega)$. Then for every $N \in \mathbb{N}$ with $N > 5$ there exists a tanh neural network $\hat{f}^N$ with two hidden layers, one of width at most $3 \lceil \frac{m+n-2}{2} \rceil \binom{m+d-1}{d} + \sum_{i=1}^{d} (b_i - a_i)(N-1)$ and another of width at most $3 \lceil \frac{d+n}{2} \rceil \binom{2d+1}{d} N^d \prod_{i=1}^{d} (b_i - a_i)$, such that for $k = 0, 1, 2$, it holds that*

$$\|f - \hat{f}^N\|_{H^k(\Omega)} \leq 2^k 3^d C_{k,m,d,f} (1+\delta) \ln^k(\beta_{k,\delta,d,f} N^{d+m+2}) N^{-m+k},$$

*where we define*

$$\beta_{k,\delta,d,f} = \frac{5 \cdot 2^{kd} \max\{\prod_{i=1}^{d}(b_i - a_i), d\} \max\{\|f\|_{W^{k,\infty}(\Omega)}, 1\}}{3^d \delta \min\{1, C_{k,m,d,f}\}},$$

$$C_{k,m,d,f} = \max_{0 \leq l \leq k} \binom{d+l-1}{l}^{\frac{1}{2}} \frac{((m-l)!)^{\frac{1}{2}}}{(\lceil \frac{m-l}{d} \rceil^{\frac{1}{2}})} \left( \frac{3\sqrt{d}}{\pi} \right)^{m-l} \|f\|_{H^m(\Omega)}.$$

*Moreover, the weights of $\hat{f}^N$ scale as $O(N \ln(N) + N^\gamma)$ with $\gamma = \frac{\max\{m^2, d(2+m+d)\}}{n}$.*

A.2 PROOF OF THEOREM 4.5

In this section, we provide the proof of Theorem 4.5. First, we need to prove the following corollary:

**Corollary A.7.** *Let $n \geq 2$, $\Omega = [0,1]^d$, $d \in \mathbb{N}_+$, $k \geq 3$. In Theorem 3.1, let $u$ be the solution of the FPE problem. Under Theorem 4.1 to Theorem 4.4, assume that $u \in H^k([0,T] \times \Omega)$, then for every $N \in \mathbb{N}$ with $N > 5$, there exist tanh neural network $u_\theta$, with two hidden layers, of widths at most $3 \lceil \frac{k+n-2}{2} \rceil \binom{d+k-1}{d} + \lceil TN \rceil + dN$ and $3 \lceil \frac{d+n+1}{2} \rceil \binom{2d+3}{d+1} TN^{d+1}$, such that*

$$L(\theta) \lesssim \tilde{C}_{k,d,T,u} \ln^4 N N^{-2k+4}, \tag{12}$$

*where $L(\theta)$ is as in equation 7, $\tilde{C}_{k,d,T,u}$ is a constant depending on $k, d, T$ and $u$.*

*Proof.* Using Theorem A.6, we have that for every $N \in \mathbb{N}$ with $N > 5$, there exist tanh neural network $u_\theta$, with two hidden layers, of widths at most $3 \lceil \frac{k+n-2}{2} \rceil \binom{d+k-1}{d} + \lceil TN \rceil + dN$ and $3 \lceil \frac{d+n+1}{2} \rceil \binom{2d+3}{d+1} TN^{d+1}$, such that for every $0 \leq l \leq 2$,

$$\|u - u_\theta\|_{H^l([0,T] \times \Omega)} \leq C_{l,k,d+1,u} \lambda_l(N) N^{-k+l},$$

where $\lambda_l(N) = 2^{l+1} 3^d (1 + \delta) \ln^l(\beta_{l,d+1,u} N^{d+k+2})$, $\delta = 1/100$, and the definition of constants are as in Theorem A.6. The weights can be bounded by $O(N^\gamma \ln N)$ where $\gamma = \max\{1, d(2 + k^2 + d)/n\}$.

Now we can bound the residuals in $L(\theta)$:

Firstly,

$$\|\frac{g^2}{2} \Delta u_\theta - \frac{g^2}{2} \Delta u\|_{L^2([0,T] \times \Omega)} \leq \frac{C_{g,max}}{2} \sqrt{d} \|u - u_\theta\|_{H^2([0,T] \times \Omega)},$$

$$\|\nabla \cdot (u_\theta \mathbf{f}) - \nabla \cdot (u\mathbf{f})\|_{L^2([0,T] \times \Omega)} \leq \|\mathbf{f}\|_{L^\infty([0,T] \times \Omega)} \|u_\theta - u\|_{H^1([0,T] \times \Omega)}$$
$$+ \|\nabla \mathbf{f}\|_{L^\infty([0,T] \times \Omega)} \|u_\theta - u\|_{H^0([0,T] \times \Omega)},$$

$$\|\partial_t u_\theta - \partial_t u\|_{L^2([0,T] \times \Omega)} \leq \|u_\theta - u\|_{H^1([0,T] \times \Omega)}.$$

So

$$\|R_{PDE}\|_{L^2([0,T] \times \Omega)} \lesssim \tilde{C}^{(1)}_{k,d,T,u} \ln^2 N N^{-k+2}.$$

One the other hand, by Theorem A.5,

$$\|\frac{g^2}{2} \nabla u_\theta - \frac{g^2}{2} \nabla u\|_{L^2([0,T] \times \partial\Omega)} \leq \frac{C_{g,max}}{2} \sqrt{\frac{2 \max\{2h_{[0,T] \times \Omega}, d\}}{p_{[0,T] \times \Omega}}} \|u_\theta - u\|_{H^2([0,T] \times \Omega)},$$

$$\|\mathbf{f} u_\theta - \mathbf{f} u\|_{L^2([0,T] \times \partial\Omega)} \leq \|\mathbf{f}\|_{L^\infty([0,T] \times \Omega)} \sqrt{\frac{2 \max\{2h_{[0,T] \times \Omega}, d\}}{p_{[0,T] \times \Omega}}} \|u_\theta - u\|_{H^1([0,T] \times \Omega)},$$

so

$$\frac{1}{T} \int_0^T \int_{\partial\Omega} |R_{bc}|^2 ds(x) dt \lesssim \tilde{C}^{(2)}_{k,d,T,u} \ln^4 N N^{-2k+4}.$$

Also using Theorem A.2 and Theorem A.5,

$$R_{ic}(\theta) = \text{KL}(p_{\text{data}}, u_\theta(\cdot, 0)) \leq \frac{1}{m} \|p_{\text{data}} - u_\theta(\cdot, 0)\|_{L^2(\Omega)} \leq \frac{1}{m} \|u_\theta - u\|_{L^2(\partial([0,T] \times \Omega))}$$

$$\leq \frac{1}{m} \sqrt{\frac{2 \max\{2h_{[0,T] \times \Omega}, d\}}{p_{[0,T] \times \Omega}}} \|u_\theta - u\|_{H^1([0,T] \times \Omega)},$$

so

$$R_{ic}(\theta) \lesssim \tilde{C}^{(3)}_{k,d,T,u} \ln N N^{-k+1}.$$

Finally, notice that $\frac{1}{T} \int_0^T \int_\Omega |R_{PDE}|^2 p_t dx dt \leq \frac{1}{mT} \|R_{PDE}\|^2_{L^2([0,T] \times \Omega)}$, combining those equations above, we can finish the proof. $\square$

*Proof of Theorem 4.5.* For $\tilde{u}$ defined in Section 3.2, denote $\hat{u} = \tilde{u} - u$. Then $\hat{u}$ satisfies the following equations:

$$\partial_t \hat{u} = \nabla \cdot (-\hat{u}\mathbf{f} + \frac{g^2}{2}\nabla \hat{u}) + R_{PDE}, \tag{13}$$

$$\hat{u}(x,0) = \tilde{u}(\cdot,0) - p_{\text{data}} =: \tilde{R}_{ic}(x), \tag{14}$$

$$\left(\hat{u}(x,t)\mathbf{f}(x,t) - \frac{g(t)^2}{2}\nabla \hat{u}(x,t)\right) \cdot \mathbf{n} = R_{bc}(\theta^\star, x, t), \forall x \in \partial\Omega. \tag{15}$$

Multiply $\hat{u}$ on both sides of Equation (13), then integration by parts and divergence theorem yields

$$
\begin{aligned}
\frac{1}{2}\frac{\partial}{\partial t}\int_\Omega |\hat{u}(x,t)|^2 \mathrm{d}x =& -\frac{1}{2}g^2 \int_\Omega \|\nabla \hat{u}\|^2 \mathrm{d}x + \frac{1}{2}g^2 \int_{\partial\Omega} (\tilde{u}(x,t) - u(x,t))(\nabla \hat{u} \cdot \mathbf{n})\mathrm{d}s(x) \\
& - \int_\Omega \hat{u}\nabla \cdot (\hat{u}\mathbf{f})\mathrm{d}x + \int_\Omega R_{PDE}\hat{u}\mathrm{d}x \\
=& -\frac{1}{2}g^2 \int_\Omega \|\nabla \hat{u}\|^2 \mathrm{d}x - \int_\Omega \hat{u}\nabla \cdot (\hat{u}\mathbf{f})\mathrm{d}x + \int_\Omega R_{PDE}\hat{u}\mathrm{d}x \\
& + \int_{\partial\Omega} (\hat{u}^2 \mathbf{f} \cdot \mathbf{n} - R_{bc}(\theta^\star, x, t))\mathrm{d}s(x) \\
=& -\frac{1}{2}g^2 \int_\Omega \|\nabla \hat{u}\|^2 \mathrm{d}x + \int_\Omega R_{PDE}\hat{u}\mathrm{d}x - \int_{\partial\Omega} R_{bc}(\theta^\star, x, t)\mathrm{d}s(x) \\
& - \int_\Omega \hat{u}\nabla \cdot (\hat{u}\mathbf{f})\mathrm{d}x + \int_\Omega \nabla \cdot (\hat{u}^2\mathbf{f})\mathrm{d}x \\
=& -\frac{1}{2}g^2 \int_\Omega \|\nabla \hat{u}\|^2 \mathrm{d}x + \int_\Omega R_{PDE}\hat{u}\mathrm{d}x - \int_{\partial\Omega} R_{bc}(\theta^\star, x, t)\mathrm{d}s(x) \\
& + \int_\Omega \hat{u}\nabla \hat{u} \cdot \mathbf{f}\mathrm{d}x \\
\leq& \frac{1}{2}(-g^2 + \|\mathbf{f}\|_\infty)\int_\Omega \|\nabla \hat{u}\|^2 \mathrm{d}x + \frac{1}{2}\int_\Omega |R_{PDE}|^2 \mathrm{d}x + \int_{\partial\Omega} |R_{bc}|\mathrm{d}s(x) \\
& + \frac{1}{2}(1 + \|\mathbf{f}\|_\infty)\int_\Omega |\hat{u}|^2 \mathrm{d}x
\end{aligned}
$$

So

$$\frac{1}{2}\frac{\partial}{\partial t}\int_\Omega |\hat{u}(x,t)|^2 \mathrm{d}x + \frac{1}{2}(g^2 - \|\mathbf{f}\|_\infty)\int_\Omega \|\nabla \hat{u}\|^2 \mathrm{d} \leq \frac{1}{2}\int_\Omega |R_{PDE}|^2 \mathrm{d}x + \int_{\partial\Omega} |R_{bc}|\mathrm{d}s(x)$$
$$+ \frac{1}{2}(1 + \|\mathbf{f}\|_\infty)\int_\Omega |\hat{u}|^2 \mathrm{d}x.$$

Integrating the above inequality over $[0,\tau]$ for $\forall 0 \leq \tau \leq T$, since $g^2 - \|\mathbf{f}\|_\infty > 0$, we obtain

$$
\begin{aligned}
& \int_\Omega |\hat{u}(x,\tau)|^2 \mathrm{d}x + \int_0^\tau (g^2 - \|\mathbf{f}\|_\infty)\int_\Omega \|\nabla \hat{u}\|^2 \mathrm{d}x\mathrm{d}t \\
\leq& \int_\Omega |\tilde{R}_{ic}(x)|^2 \mathrm{d}x + (1 + \|\mathbf{f}\|_\infty)\int_0^\tau \int_\Omega |\hat{u}(x,t)|^2 \mathrm{d}x\mathrm{d}t \\
& + 2\int_0^\tau \int_{\partial\Omega} |R_{bc}(\theta^\star, x, t)|\mathrm{d}s(x)\mathrm{d}t + \int_0^\tau \int_\Omega |R_{PDE}|^2 \mathrm{d}x\mathrm{d}t. \\
\leq& (1 + \|\mathbf{f}\|_\infty)\left(\int_0^\tau \int_\Omega |\hat{u}(x,t)|^2 \mathrm{d}x\mathrm{d}t + \int_0^\tau \int_0^t (g^2 - \|\mathbf{f}\|_\infty)\int_\Omega \|\nabla \hat{u}(x,s)\|^2 \mathrm{d}x\mathrm{d}s\mathrm{d}t\right) \\
& \int_\Omega |\tilde{R}_{ic}(x)|^2 \mathrm{d}x + 2\int_0^\tau \int_{\partial\Omega} |R_{bc}(\theta^\star, x, t)|\mathrm{d}s(x)\mathrm{d}t + \int_0^\tau \int_\Omega |R_{PDE}|^2 \mathrm{d}x\mathrm{d}t.
\end{aligned}
$$

Using Grönwall's inequality, we have

$$\int_\Omega |\hat{u}(x,\tau)|^2 \mathrm{d}x + \int_0^\tau (g^2 - \|\mathbf{f}\|_\infty) \int_\Omega \|\nabla \hat{u}\|^2 \mathrm{d}x \mathrm{d}t$$

$$\leq \left(1 + (1+\|\mathbf{f}\|_\infty)Te^{(1+\|\mathbf{f}\|_\infty)T}\right)$$

$$\left(T \int_\Omega |\tilde{R}_{ic}(x)|^2 \mathrm{d}x + \int_0^T \int_\Omega |R_{PDE}(\theta^\star, x, t)|^2 \mathrm{d}x \mathrm{d}t + 2 \int_0^T \int_\Omega |R_{bc}(\theta^\star, x, t)| \mathrm{d}s(x)\mathrm{d}t\right)$$

$$\leq \left(1 + (1+\|\mathbf{f}\|_\infty)Te^{(1+\|\mathbf{f}\|_\infty)T}\right)$$

$$\left(2MT \cdot \mathrm{TV}(p_{\text{data}}, \tilde{u}(\cdot,0)) + \int_0^T \int_\Omega |R_{PDE}(\theta^\star, x, t)|^2 \mathrm{d}x \mathrm{d}t + 2 \int_0^T \int_\Omega |R_{bc}(\theta^\star, x, t)| \mathrm{d}s(x)\mathrm{d}t\right)$$

$$\leq \left(1 + (1+\|\mathbf{f}\|_\infty)Te^{(1+\|\mathbf{f}\|_\infty)T}\right)$$

$$\left(\sqrt{2}MT\sqrt{\mathrm{KL}(p_{\text{data}}, \tilde{u}(\cdot,0))} + \int_0^T \int_\Omega |R_{PDE}(\theta^\star, x, t)|^2 \mathrm{d}x \mathrm{d}t + 2 \int_0^T \int_\Omega |R_{bc}(\theta^\star, x, t)| \mathrm{d}s(x)\mathrm{d}t\right)$$

$$\leq \left(1 + (1+\|\mathbf{f}\|_\infty)Te^{(1+\|\mathbf{f}\|_\infty)T}\right)$$

$$\left(\sqrt{2}MT\sqrt{R_{ic}(\theta^\star)} + \int_0^T \int_\Omega |R_{PDE}(\theta^\star, x, t)|^2 \mathrm{d}x \mathrm{d}t + 2 \int_0^T \int_\Omega |R_{bc}(\theta^\star, x, t)| \mathrm{d}s(x)\mathrm{d}t\right)$$

$$\lesssim C_1 T^2 e^{C_2 T} \left(L(\theta^\star)^{1/2} + L(\theta^\star)\right)$$

For $\forall \theta \in \Theta$, by the convergence of Monte-Carlo integrals, we have that

$$\left|\int_0^T \int_{\partial\Omega} |R_{bc}|^2 \mathrm{d}s(x)\mathrm{d}t - \frac{1}{N_{bc}N_T} \sum_{k=1}^{N_T} \sum_{i=1}^{N_{bc}} |R_{bc}(\theta, x_{bc}^{(k,i)}, t_k^{bc})|^2\right| \lesssim (N_{bc}N_T)^{-1/2}.$$

On the other hand, since for $\forall k \in \{1, \cdots, N_T\}$, the samples in $\{x_{PDE}^{(k,i)}\}_{i=1}^{N_{PDE}}$ are all i.i.d, and $\hat{u}$, $\partial_t \hat{u}$, $\|\nabla \hat{u}\|$, $\Delta \hat{u}$, $\mathbf{f}$ and $g$ are all bounded functions, we have

$$\left|\int_\Omega |R_{PDE}|^2 p_{t_k}(x)\mathrm{d}x - \frac{1}{N_{PDE}} \sum_{i=1}^{N_{PDE}} |R_{PDE}(\theta, x_{PDE}^{(k,i)}, t_k)|^2\right| \lesssim N_{PDE}^{-\frac{1}{2}}$$

with probability at least $1 - \frac{N_T}{N_{PDE}^{\frac{1}{2}}}$ uniformly for all $k = 1, \cdots, N_T$.

Consequently,

$$\left|\frac{1}{T}\int_0^T \int_\Omega |R_{PDE}|^2 p_{t_k}(x)\mathrm{d}x\mathrm{d}t - \frac{1}{N_T}\frac{1}{N_{PDE}} \sum_{k=1}^{N_T} \sum_{i=1}^{N_{PDE}} |R_{PDE}(\theta, x_{PDE}^{(k,i)}, t_k)|^2\right| \lesssim (N_{PDE}N_T)^{-\frac{1}{2}}$$

with probability at least $1 - \frac{N_T}{\sqrt{N_{PDE}}}$.

Similarly, we have

$$\left|R_{ic}(\theta) - \left(\int_\Omega p_{\text{data}}(x)\log(p_{\text{data}}(x))\mathrm{d}x - \frac{1}{N_{tb}} \sum_{i=1}^{N_{tb}} \log(u_\theta(x_{tb}^{(i)}, 0))\right)\right| \lesssim N_{tb}^{-\frac{1}{2}}$$

with probability at least $1 - \frac{1}{\sqrt{N_{tb}}}$.

Consider $\theta_0$ such that $u_{\theta_0}$ satisfies Equation (12) (the existence of $\theta_0$ is guaranteed by Theorem A.7), since $\tilde{u} = u_{\theta^\star}$ is the minimizer of $\hat{L}_{train}$ (thus the minimizer of $L_{train}$), we have $L_{train}(\theta^\star) \leq L_{train}(\theta_0)$. Concluding from above, we have that with probability at least $1 - \frac{2N_T}{\sqrt{N_{PDE}}} - \frac{2}{\sqrt{N_{tb}}}$,

$$|L_{train}(\theta^\star) - L(\theta^\star)| \lesssim (N_{bc} + N_{PDE} + N_{tb})^{-1/2} N_T^{-1/2},$$
$$|L_{train}(\theta_0) - L(\theta_0)| \lesssim (N_{bc} + N_{PDE} + N_{tb})^{-1/2} N_T^{-1/2}.$$

As a result, with probability at least $1 - \frac{2N_T}{\sqrt{N_{PDE}}} - \frac{2}{\sqrt{N_{tb}}}$,

$$\int_\Omega |\hat{u}(x,\tau)|^2 \mathrm{d}x + \int_0^\tau (g^2 - \|\mathbf{f}\|_\infty) \int_\Omega \|\nabla \hat{u}\|^2 \mathrm{d}x \mathrm{d}t$$

$$\lesssim C_1 T^2 e^{C_2 T} \left( L(\theta^\star)^{1/2} + L(\theta^\star) \right)$$

$$\lesssim C_1 T e^{C_2 T} \left( L_{train}(\theta^\star)^{1/2} + L_{train}(\theta^\star) + 2(N_{bc} + N_{PDE} + N_{tb})^{-1/4} N_T^{-1/4} \right)$$

$$\lesssim C_1 T^2 e^{C_2 T} \left( L_{train}(\theta_0)^{1/2} + L_{train}(\theta_0) + 2(N_{bc} + N_{PDE} + N_{tb})^{-1/4} N_T^{-1/4} \right)$$

$$\lesssim C_1 T^2 e^{C_2 T} \left( L(\theta_0)^{1/2} + L(\theta_0) + 4(N_{bc} + N_{PDE} + N_{tb})^{-1/4} N_T^{-1/4} \right)$$

$$\lesssim C_1 T^2 e^{C_2 T} \left( \sqrt{\tilde{C}_{k,d,T,u}} \ln^2 N N^{-k+2} + (N_{bc} + N_{PDE} + N_{tb})^{-1/4} N_T^{-1/4} \right).$$

Finally, since $\min_{t \in [0,T]}(g(t)^2 - \|\mathbf{f}\|_\infty) = \delta_0 > 0$, we have that for $\forall t \in [0,T]$, with probability at least $1 - \frac{2N_T}{\sqrt{N_{PDE}}} - \frac{2}{\sqrt{N_{tb}}}$,

$$\|\hat{s}_t - \nabla \log p_t\|_{L^2([0,t] \times \Omega)} = \left( \int_0^t \int_\Omega \| \frac{\nabla u_{\theta^\star}(x,\tau)}{u_{\theta^\star}(x,\tau)} - \frac{\nabla u(x,\tau)}{u(x,\tau)} \|^2 \mathrm{d}x \mathrm{d}\tau \right)^{1/2}$$

$$\lesssim \frac{M}{m^2} \left( \int_0^t \int_\Omega |\hat{u}(x,\tau)|^2 \mathrm{d}x \mathrm{d}\tau + \int_0^t \int_\Omega \|\nabla \hat{u}\|^2 \mathrm{d}x \mathrm{d}\tau \right)^{1/2}$$

$$\leq \frac{M \sqrt{\min\{T, 1/\delta_0\}}}{m^2} \left( \int_\Omega |\hat{u}(x,t)|^2 \mathrm{d}x + \int_0^t (g(\tau)^2 - \|\mathbf{f}\|_\infty) \int_\Omega \|\nabla \hat{u}(x,\tau)\|^2 \mathrm{d}x \mathrm{d}\tau \right)^{1/2}$$

$$\lesssim \frac{M \sqrt{\min\{T, 1/\delta_0\}}}{m^2} \sqrt{C_1 T^2 e^{C_2 T}} \left( (\tilde{C}_{k,d,T,u})^{1/4} \ln N N^{-\frac{k}{2}+1} + (N_{bc} + N_{PDE} + N_{tb})^{-1/8} N_T^{-1/8} \right)$$

$$\lesssim C \left( (\tilde{C}_{k,d,T,u})^{1/4} \ln N N^{-\frac{k}{2}+1} + (N_{bc} + N_{PDE} + N_{tb})^{-1/8} N_T^{-1/8} \right)$$

Let $\epsilon_0 = (\tilde{C}_{k,d,T,u})^{1/4} \ln N N^{-\frac{k}{2}+1}$, then $\epsilon_0$ can be arbitrary small by choosing large enough $N$ (i.e. the width of the training neural networks), thus proving the desired error bounds. $\square$

