# OpenReview forum: "Solving Reflected Diffusion Models: A PINN-based Method"
_ICLR.cc/2026/Conference — ICLR 2026 Conference Withdrawn Submission_

### Official Review · Reviewer_zeMw · 2025-10-30

**Soundness:** 1
**Presentation:** 1
**Contribution:** 2
**Rating:** 2
**Confidence:** 4

**Summary:**

This paper introduces a score estimator using Physics-Informed Neural Network (PINN) to solve reflected diffusion models, which aims to fill the gap and provide a score-estimation methodology of the score estimator in the reflected diffusion setting with theoretical guarantees. This paper claims that they obtain a uniform-in-time score estimation error for reflected diffusion models requiring smoothness and boundness of the data distribution, in terms of Fokker-Planck equation associated with reflected stochastic differential equation (PDE).

**Strengths:**

This paper focuses on theoretical proof, which shows the existence of an optimal minimizer with a uniform-in-time error bound for score estimation which is dimension independent and this guarantees the performance of the estimator for high-dimensional generative modeling.

**Weaknesses:**

1. Notation is a mess to me: the drift term is written in bold f, and the function in the function class F is also with notation f; the first time u appears in 2.3 without introducing what it is, not until in theorem 3.1, it is assigned as the density function.
2. In the algorithm, it briefly says we can sample interior training set from data distribution and boundary training set from the boundary set. Interior training set might be a comparatively easy thing to do, but how to sample the boundary set? Is there always a function to check if it is on the boundary? What's the computation cost here? How large are the interior and boundary training sets are supposed to be? Is this realistic to sample in the real-word applications? The algorithm is very unclear to me and lack of discussion.
3. No empirical evidence to show the proposed algorithm works. No experiment is done.
4. This paper is limited to discussions to compare with other generative models, the most direct one I can think of is simply adding a boundary constraints on score matching loss.

**Questions:**

Following the weaknesses above,

1. Since the paper starts with introducing the reflected diffusion model, it is reasonable to try all the experiments done in reflected diffusion model and compare the results to show the proposed method works.
2. Will the computation cost be high in high-dimensional PDEs?
3. Again no experiment to show the evidence that the uniform theoretical error bound is valid. Do you think there could be a discrepancy between this theoretical analysis will be aligned with the experimental results?
4. Since the theorem requires smoothness and boundness, what if the data distribution in the real-word application does not satisfy these assumptions? Can you loose these conditions a bit?

---

### Official Review · Reviewer_tQRZ · 2025-10-31

**Soundness:** 2
**Presentation:** 2
**Contribution:** 2
**Rating:** 2
**Confidence:** 3

**Summary:**

This paper studies score matching for reflected diffusion models and proposed to use Physics-Informed Nueral Network to solve PDE instead of standard methods like denoising score matching in bounded domains. The main motivation is to ensure a uniform bound of score approximation error by solving the PDEs using neural nework. Theorectical results are provided that this will ensure a desired convergence to the data distribution.

**Strengths:**

The theoretical motivation is valid and sound. It will be nice to have theorectical guarantees that trained scores will have bounded errors, so that theoretical guarantees are ensured.

**Weaknesses:**

There are two major issues that remain in this paper.
1. First, there is no comparison to or illustration of the standard way of learning the scores of discrete diffusion models by denosing score matching or bounded denoising score matching in Lou&Ermon. The proposed PINN introduced extra complexity in learning the score, yet there is no theoretical evidence in such benefits over the baseline approach.
2. There lacks experiments to showcase the benefits and scalability of the proposed methods. Although this is a theoretical oriented part, there is no experiments in showcasing the convergence of the algorithms under simple 2d data settings, not to say if the algorithms are scalable and can efficiently learn the scores in high dimension cases. Theorectical bounds are not sufficient or convincing enough to showcase the benefits of the proposed approach.

**Questions:**

1. What is the algorithmic complexity of learning the score use PINN compared to standard DSM techniques?
2. How will the convergence to the optimum go in pratice in e.g. 2-dim bounded datasets or image datasets? Can PINN learned scores showcase better loss compared to bounded DSM setting?

---

### Official Review · Reviewer_h2TP · 2025-10-31

**Soundness:** 3
**Presentation:** 2
**Contribution:** 2
**Rating:** 4
**Confidence:** 3

**Summary:**

The authors present an analysis on reflected diffusion models, and attempt to quantify error bounds in terms of sample number. They attempt this by analyzing the score based SDE throught the Fokker-Planck equation and applying a physics informed neural network (PINN) to solve it. The authors analyze the performance of the resulting PINN based estimator to arrive at the main result of the paper, which is an error bound on the score approximator in terms of the number of samples.

**Strengths:**

- Reflected diffusion is a useful tool and understanding where it's limitations are and how to improve it is an important topic.
- Solving the problem through the PDE and using tools from that world is an interesting approach.

**Weaknesses:**

- The notation in the paper is messy to the point where it is difficult to follow, for example: in the un-numbered equations around line 275, there is an "f", but later there is an f0, but also an f0(x), f(vec(x),vec(t)), and later an f^(j)_0(x). As far as I can tell, there is no no vec(t) defined, and the parenthesized (j) superscript in f^(j)_0(x) is also not defined
- My main grip with the paper is that the main result of theorem 4.5 seems to not have anything to do with reflected diffusion in particular. Why is it that the authors choose to address reflected diffusion in particular?
- Following on that, theorem 4.7 is cited from another paper, and while the authors suggest how to combine it with their own work, they don't actually do that.
- The conclusion that the error bound goes as O(N-1/4) seems to be incorrect, since it has nothing to do with the dataset size N, but rather seems to depend on the number of discretization points of the PINN. Secondly, I struggle to see how the factors theorem 4.5 grow as N^-1/4. Thirdly, there is a constant term epsilon_0 in the error bound that makes scaling at large N irrelevant, so it would rather go as O(1). Depending on the actual values of epsilon_0 and the magnitude of N, epsilon_0 could be an important term.
- If the relation the authors develop is truly only dependent on the number of quadrature points in the PINN, they should defend more strongly how this result generalizes in the absence of PINNs, for "regular" reflected diffusion models.
- I find it a missed opportunity there is no experiments that demonstrate the method, I think an experiment in a theoretical paper can make it much more approachable for a wider audience
- There are some scattered grammar issues throughout the text, I would recommend the authors use some language model to address those.

**Questions:**

- see also weaknesses
- Can the authors clarify what all the N factors are in thm 4.5, and if any them actually relate to "dataset size" as they state in the abstract?
- Could the authors clarify "it is not hard to see" under remark 3.2? Is this just a solution to the PDE? If it is, it would be worth adding an appendix with a derivation. In particular, it is difficult to understand what the conditions have to do with the solution specifically.
- The authors define the SDE between 0 and T, (Thm 3.2) but the stationary distribution (Remark 3.2) seems to be defined at infinity. Can the authors clarify these boundary conditions? It seems to me that the setup on both sides does not quite match. These may be technical details but they should be addressed.

---

### Official Review · Reviewer_18xm · 2025-11-01

**Soundness:** 2
**Presentation:** 2
**Contribution:** 2
**Rating:** 4
**Confidence:** 3

**Summary:**

This paper proposes a score estimator using the Physics-Informed Neural Network (PINN), solving reflected diffusion models in a deep-learning fashion. It proves a uniform theoretical error bound of O($N^{-0.25}$ ) for the score function on a training dataset of sample size N at any time $t \in [0, T]$ of the diffusion process, which is dimension independent.

**Strengths:**

Theorem 4.5. is valuable form a theoretical point of view.

The paper presents the background material quite well facilitating the understanding of the novel results.

**Weaknesses:**

**W1** The main weakness is that the paper does not test the proposed estimator in practice. Given the ICLR preferred format and the focus on impact, a strong experimental section would have boosted the paper.

**W2** A strong purely-theoretical paper would have been fine if it represented 'deep' or multiple valuable results. However, the only valuable contribution in the current submission is Theorem 4.5. which while valuable, is quite straightforward logically. It provides a bound on the convergence speed, but the convergence itself is clear, since as long as $R_{PDE}$, $R_{bc}$ and $R_{ic}$ are completely minimized, then the estimator will match the real score and satisfy boundaries, making the KL in Equation (11) zero. Thus, while for the sake of rigor it is important to derive lines 870, 871, such a conclusion is expected.

**W3** Algorithm 1 which is not tested in practice, seems to indicate significant increase in compute resources and time. Without testing it on real world data, it is unclear how effective and efficient it is.

**Questions:**

The derived bound hold with probability $1-\frac{2N_T}{\sqrt{N_{PDE}}}-\frac{2}{\sqrt{N_{tb}}}$. What are the implications regarding the needed growth of interior points vs time-points ?

---

### Note · Authors · 2025-12-03

I have read and agree with the venue's withdrawal policy on behalf of myself and my co-authors.